# Exploiting radar polarimetry for nowcasting thunderstorm hazards using deep learning

Nathalie Rombeek[1], Jussi Leinonen[1], and Ulrich Hamann[1]

[1]Federal Office of Meteorology and Climatology MeteoSwiss, Locarno-Monti, Switzerland

**Correspondence:** Ulrich Hamann (ulrich.hamann@meteoswiss.ch)

**Abstract.** This work presents the importance of polarimetric variables as an additional data source for nowcasting thunderstorm hazards using an existing neural network architecture with recurrent-convolutional layers. The model can be trained to predict different target variables, which enables nowcasting of hail, lightning and heavy rainfall for lead times up to 60 min with a 5 min resolution, in particular. the exceedance probabilities of Swiss thunderstorm warning thresholds predicted. This study is based on observations from the Swiss operational radar network, which consists of five operational polarimetric C-band radars. The study area of the Alpine region is topographically complex and has a comparatively very high thunderstorm activity. Different model runs using combinations of single- and dual-polarimetric radar observations and radar quality indices are compared to the reference run using only single-polarimetric observations. Two case studies illustrate the performance difference when using all predictors compared to the reference model. The importances of the predictors are quantified by investigating the final training loss of the model, skill scores such as CSI, precision, recall, precision-recall area-under-the-curve, and the Shapley value. Results indicate that single-polarization radar data is the most important data source. Adding polarimetric observations improves the model performance compared to reference model in term of the training loss for all three target variables. Adding quality indices does so, too. Including both, polarimetric variables and quality indices at the same time improves the accuracy of nowcasting heavy precipitation and lightning, with the largest improvement found for heavy precipitation. No improvement could be achieved for nowcasting of the probability of hail in this way.

## 1 Introduction

Severe convective weather events, such as hail, lightning and heavy precipitation, are likely to increase across Europe during this century (Rädler et al., 2019; Raupach et al., 2021; Taszarek et al., 2021). The heavy rainfall associated with these convective storms can turn into flash floods and land slides and, consequently, be a great threat to human lives (Holle et al., 1993; Lynn and Yair, 2010). Additionally, a considerable part of the total weather-related economic losses are caused by severe convective weather (Hoeppe, 2016). Therefore, accurate short-term predictions of convective events are of interest, as they allow to issue warnings in order to reduce societal and economic impact.

Stratiform precipitation typically has larger spatial scales and last longer than severe convection. Numerical weather prediction (NWP) models are particularly suited for this purpose. On the other hand, simulating severe convection with its short time and spatial scales, for which the exploitation of the most recent observations is essential, is very challenging. Accordingly,

many weather services aim for rapid update cycles, i.e. an hourly instead of the former three-hour update cycle. Due to the computational demand of the assimilation and prediction, more frequent update cycles than hourly are currently not feasible. The results of NWP models are typically available after several tens of minutes (e.g. a COSMO-1E run requires 50 minutes runtime). NWP analysis is a combination of previous model predictions and the latest available observations, and the assimilation creates a physically consistent state of the atmosphere, which typically deviates slightly from the latest observations. Meanwhile, nowcasting algorithms aim to provide their output within tens of seconds up to a minute (Pierce et al., 2012). They typically do not strive for a physically consistent representation of the atmosphere, but do make use of the latest observations, which results in higher performance on the very short and short time scales (i.e. 1 h) and smaller scales (Simonin et al., 2017) (but inferior performance on longer lead times). As localized warnings are often issued for the very short term time range, nowcasting plays a crucial role in warning systems for severe convection.

Weather radars are often utilized for nowcasting purposes as they provide near real time input data with a high resolution and broad spatial coverage. Conventional nowcasting techniques typically extrapolate the latest observations from weather radars in time, based on either estimation of the motion field such as used in Pysteps (Pulkkinen et al., 2019), NowPrecip (Sideris et al., 2020) or rainymotion (Ayzel et al., 2019), or identifying and tracking individual storms, e.g. the Thunderstorms Radar Tracking algorithm (TRT; Hering et al., 2004) or Thunderstorm Identification, Tracking, Analysis, and Nowcasting (TITAN; Dixon and Wiener, 1993). However, these methods often have difficulties to take the life cycle of convective cells with growth and dissipation processes into account and consequently result in relatively short skilful lead times for convective weather (Imhoff et al., 2020; Foresti et al., 2016; Wilson et al., 1998).

In recent years, there have been significant advances in using deep learning for generating nowcasts of heavy precipitation using radar as input, e.g. Guastavino et al. (2022), Han et al. (2021), Ritvanen et al. (2023) and Yin et al. (2021), or in the case of Leinonen et al. (2023) including multiple data sources. In addition, radar is also exploited as predictor for nowcasting lightning e.g. by Leinonen et al. (2022b) and Zhou et al. (2020). However, these studies primarily focus on single-polarization radar observations (e.g. precipitation rates based on horizontal reflectivity or the reflectivity itself), and do not utilize polarimetry explicitly, despite polarimetry can provide further information about the micro-physical properties of hydrometeors. Hence, adding polarimetric radar variables explicitly helps considerably to reduce ambiguities concerning the hydrometeor classes and drop size distributions.

Dual-polarization radars have two orthogonally polarized beams, making it possible to derive additional properties such as particle shape and to some extent the size, which are useful for meteorological applications (Fabry, 2018; Kumjian, 2013b). Hydrometeor classification algorithms such as those developed by Besic et al. (2016) and Vivekanandan et al. (1999) use this extra information to identify different hydrometeors. Other studies showed the potential of polarimetric variables for providing information on other convective hazards, such as hail and lightning (Figueras i Ventura et al., 2019; Lund et al., 2009), or the evolution of convective storms (Snyder et al., 2015). However, interpretation of polarimetric signatures for convective weather forecasts remains challenging (Kumjian, 2013a, b), and requires more advanced data processing techniques such as machine learning.

This research investigates the additional value of polarimetric variables for nowcasting severe convective weather, which includes hail, lightning and severe precipitation. Data source importance is explored by performing both a qualitative and quantitative analysis (i.e. focal loss or cross entropy, Shapley values, critical success index and fractions skill score). We use the recurrent-convolutional deep learning model from Leinonen et al. (2023), as it is able to utilize multiple data sources and predict, with a slight modification, multiple extreme events.

One of the first successful attempts to incorporate polarimetric variables for nowcasting convective precipitation using deep learning was realized by Pan et al. (2021). However, that study exploits only observations in 3 km altitude, i.e. the Constant Altitude Plan Projection Indicator (CAPPI). In this study, we exploit relevant hydrometeors and their characteristics from multiple altitudes. In addition, we investigate the potential to nowcast not only precipitation, but also hail and lightning, by utilizing polarimetric variables.

This paper introduces the data used for training in section 2, while section 3 describes the model architecture. Results are described and discussed in section 4, and section 5 concludes the article.

## 2   Data

For training purposes, the "precipitation radar" dataset from Leinonen et al. (2022b) was used (here named "Single-pol Radar"), which is described in more detail in the corresponding paragraph below. The training dataset was extended with polarimetric
variables retrieved from the Swiss operational radar network, and quality indices from Feldmann et al. (2021). The data were collected from April to September 2020. The creation of training samples is described in more detail in section 3.1.

### 2.1   Operational radar network

The study area is completely covered by the Swiss operational radar network, which consists of five operational polarimetric C-band radars (Germann et al., 2022). Operationally available products have a resolution of 500 m, comprising 20 elevation
scans from -0.2° to 40° within 5 min per radar. The maximum observation range of a single radar is 246 km. In total, the study area covers more than 400.000 $km^2$.

     A sophisticated data-processing chain including bias correction, removal of ground clutter and non-weather echoes, visibility correction and vertical profile correction (Germann et al., 2006) retrieves a high quality, radar-based precipitation estimate at the surface (RZC). The final radar products that are used as input for the deep learning algorithm have a resolution of 1 km.

### 2.2   Data sources and processing

The model was trained based on all possible combinations of the three data sources below:

     **(1) Single-pol radar** (R) data retrieved from the Swiss operational weather radar network (Germann et al., 2022). The considered variables in this source are the rain rate at the surface, column maximum echo intensity and altitude, echo top height at radar reflectivity thresholds of 20 and 45 dBZ, and the vertically integrated water content. This source was used and

described in more detail in Leinonen et al. (2022b). Note that dual-polarization data were used for clutter suppression in the processing chain of the Swiss operational weather radar network.

**(2) Polarimetric variables** (P) also obtained from the Swiss operational radar network. The considered polarimetric variables in this research are the reflectivity factor at vertical polarization ($Z_V$), differential reflectivity ($Z_{dr}$), co-polar cross-correlation coefficient ($\rho_{hv}$), and specific differential phase ($K_{dp}$).

$Z_{dr}$ is an indicator for shape, with positive values indicating targets larger in the horizontal than the vertical dimension. Such targets include large raindrops, which are flattened by aerodynamic forces while falling, but not solid hailstones, which tend to be round and therefore have values close to 0 (Seliga and Bringi, 1976).

$K_{dp}$ is an indication for concentration and shape, and is used as a measure for rain intensity (Sachidananda and Zrnic, 1986). Positive values can be an indicator for heavy rain, while negative values means that targets are more elongated vertically than horizontally (e.g. graupel) and values close to zero indicate nearly round or randomly oriented particles (Rinehart, 2010). One advantage of $K_{dp}$ over $Z_{dr}$ is that it is unaffected by differential attenuation.

$\rho_{hv}$ indicates homogeneity, with smaller values indicating more heterogeneity among the shape, size and orientation of the detected particles (Fabry, 2018).

To reduce the dimensionality and estimate values at the ground level, the polarimetric variables at various altitudes are aggregated following the method of Wolfensberger et al. (2021), a weighted sum taking both static radar visibility and height above the ground level of each point into account. Radar visibility is determined by the fraction of the radar beam that is not blocked due to partial and total beam shielding by the complex mountainous terrain. The weight is determined using a linear relationship with visibility and an exponential relationship with height:

$$w(h) = \exp(\beta \frac{h}{1000}) \frac{\text{VIS}}{100} \tag{1}$$

Here, $h$ represents the height above the ground of the observation in meters, $\beta$ $(m^{-1})$ is the slope of the exponent, and VIS is the visibility (%). A sensitivity study showed that a value of –0.5 for $\beta$ is best suited for precipitation retrieval (Wolfensberger et al., 2021), consequently, the same value is used here. First, the polarimetric data were transformed by normalizing the standard deviation and by shifting the mean to 1. Second, to reduce presence of noise, fields were compared with RZC, and set to zero where RZC does not contain precipitation.

**(3) Quality indices** (Q) obtained from Feldmann et al. (2021). Quality of radar observations in mountainous terrain fluctuates over elevations and is influenced by the scanning strategy. Especially at low levels, visibility is reduced as a consequence of radar beam blockage. The quality of the observations at every location is influenced by multiple properties. The quality index combines the following factors into a single index: visibility, minimum altitude of observation, maximum altitude of observation and numerical noise.

## 2.3 Targets

The same target variables from Leinonen et al. (2023) and Leinonen et al. (2022b) are derived:

**Lightning occurrence** is obtained from the observations by the EUCLID lightning network (Schulz et al., 2016; Poelman et al., 2016), delivered to MeteoSwiss by Météorage. The point-data were transformed to a gridded binary map, with 1 indicating lightning activity within a radius of 8 km and in the last 10 min, and 0 otherwise. This definition is used in safety procedures at airports for takeoff and landing operations, and based on the regulations of the European Union (2017) and International Civil Aviation Organization (2018). In this way, the result of our machine learning algorithm can be directly applied for METAR trend reports without any adjustment of the temporal and spatial resolution.

**Probability of hail (POH)** is the probability of hail reaching the ground. This a product from the operational MeteoSwiss radar network, using the formula from Foote et al. (2005) based on Waldvogel et al. (1979). It utilizes the difference between the 45 dBZ echo top level and the freezing level.

**CombiPrecip** is an operational product of MeteoSwiss for precipitation combining real-time radar and rain-gauge observations to adjust the biases often are observed in radar measurements (Sideris et al., 2014a, b). The CombiPrecip product is converted into a probability distribution. CombiPrecip estimations are considered as the expected precipitation $\mathrm{E}[R]$. The standard deviation is approximated as $\mathrm{Std}[R] = 0.33\,\mathrm{E}[R]$ by separating the error due to the lack of rain gauge representation from the uncertainty in the radar measurement, using the method from Ciach and Krajewski (1999). The probability distribution is transformed to probabilistic estimates for four precipitation classes, based on warning levels of MeteoSwiss. The thresholds are $R_0 = 0$, $R_1 = 10\,\mathrm{mm}$, $R_2 = 30\,\mathrm{mm}$ and $R_3 = 50\,\mathrm{mm}$ precipitation aggregated over 60 min at a 1 km$^2$ grid point. Probabilities $q_c$ are assigned to each class $c \in [0,3]$ as

$$
q_c = \int\limits_{R_c}^{R_{c+1}} p(R)\,\mathrm{d}R \tag{2}
$$

where $p$ is a lognormal probability distribution function. Note that the machine learning model can be adapted to calculate a larger number of thresholds. In this publications, we concentrate on these four thresholds representing the warning levels of MeteoSwiss.

## 3  Methods

### 3.1  Event selection

The radar-derived rainfall rate was used to select training samples where convective activity was likely to happen. Regions with 10 neighbouring pixels that exceeded 10 mm h$^{-1}$ were located, and at every timestep for $\pm 2\,h$ a box of $256 \times 256$ km$^2$ was added to the identified region. Duplicated regions were removed by dividing the study area into tiles of $32 \times 32$ km$^2$, and storing only unique tiles that do not overlap in time and space simultaneously. This resulted in a total of $30\,641$ different starting times for the training sequences. In total $1\,021\,447$ different samples could be created in this way (not including the further diversity added by data augmentation). Around 10 % of the total training samples was used for validation, another 10 % for testing and the rest for training. Entire days were randomly selected for either the validation, testing or training set to minimize

the correlation between the datasets. The event selection process is identical to that of Leinonen et al. (2022b); more detailed description of the selection procedure can be found in that article.

## 3.2 Neural network

The recurrent-convolutional deep learning model from Leinonen et al. (2023) is used, adding the newly introduced sources described in Sect. 2.2. The recurrent connections enable to model the temporal evolution, while the convolutional connections model the spatial structure. This model has an encoder-forecaster framework, in which the encoder produces a deep representation of the atmospheric state, which is decoded into a prediction by the forecaster. It has a generic architecture, making it possible to predict lightning, POH and heavy precipitation by only changing the target. The main difference between the predicted thunderstorm hazards is that the output of heavy precipitation is accumulated over 1 hour for predefined warning levels, whereas hail and lightning are produced at a 5-min resolution for 12 time steps (1 hour). In order to make the results comparable with (Leinonen et al., 2023), a maximum lead time of 60 min is selected here. For a more detailed description of the model we refer to the publications of Leinonen et al. (2022b) and Leinonen et al. (2023).

Hail and precipitation targets have a probabilistic output, and for that reason cross entropy (CE; Goodfellow et al., 2016) was used as a loss function. CE measures the difference in the probability distributions between the true distribution and predicted distribution of the target classes. To be consistent with Leinonen et al. (2023), the focal loss (Lin et al., 2017) is used for lightning. The focal loss is an adaptation of the CE and focuses more on the pixels whose classification is more uncertain ($p_t < 0.5$). In which $p_t$ is the predicted probability of the target.

In order to estimate the influence of the random weight initialization on the consistency of the model, we trained the model with each possible combination of data sources three times. As the sample size is rather small, we used the unbiased sample standard deviation for calculating the standard deviation between these runs.

In order to have variation in the training process and reduce overlap, during each epoch one training sample is randomly selected for each starting time. For the validation set, a fixed set of samples was used to compute the validation loss after each epoch in order to avoid coincidental improvement in the loss. The number of epochs was not fixed; instead, an early stopping strategy was employed. The learning rate is divided by 5 when the loss in the validation set has not improved for three consecutive epochs, and the training ends when the loss in the validation set did not improve for six consecutive epochs. The weights corresponding to the best validation loss are saved in the end. On average training stopped after 20–30 epochs, for which one epoch took around 18 min of time on a computing cluster node with eight Nvidia V100 GPUs.

Contrary to the training time, it takes only 8 seconds to nowcast one hazard with 12 time steps on a machine with 4 CPUs (Intel(R) Xeon(R) Gold 6142 CPU @ 2.60GHz), requiring 16 GB of RAM.

## 3.3 Importance of data sources

The importance from individual data sources can be assessed using the Shapley value (Shapley, 1951) as a quantitative indicator of the total importance of each data source. The total contribution among the predictors is distributed by assigning a value that represents their marginal contribution. For more information on calculating the Shapley value we refer to the description of

Molnar (2022) (chapter 9.5). We normalize the sum of the values of the individual components to add up to 1, with higher values indicating higher importance.

## 3.4 Model evaluation

Before calculating different metrics to evaluate the models, the target variables for hail and precipitation were transformed to binary fields. For hail a threshold of $0.5$ was selected, meaning that a $POH \geq 50\%$ is considered as hail and set to 1, otherwise 0. For precipitation the skill score per class is analysed by summing all probabilities in and above the selected class. Second, a threshold of $0.5$ is used, with setting probabilities $\geq 0.5$ to 1.

The models are evaluated based on the critical success index (CSI), precision recall (PR) curve and the fractions skill score (FSS). The CSI and PR curve are based on contingency tables, containing true positives (TP), false positives (FP), false negatives (FN) and true negatives (TN).

CSI indicates the amount of events that were correctly predicted:

$$CSI = \frac{TP}{TP + FP + FN}. \tag{3}$$

When there is an imbalance between two classes (no event and event), the PR curve is a useful tool for interpretation of probabilistic forecasts. Precision indicates how good the model is at predicting an event:

$$Precision = \frac{TP}{TP + FP} \tag{4}$$

Recall gives the fraction of events that were predicted:

$$Recall = \frac{TP}{TP + FN} \tag{5}$$

The PR curve is obtained by computing both precision and recall at all threshold levels ranging from 0 to 1. The information of the PR-curve can be summarized by the area under the curve (AUC). A larger AUC indicates a better-performing model over the whole range of thresholds.

The FSS is a measure for neighbourhood verification, which measures the skill of the forecast in predicting the occurrence of an event at a selected spatial scale (Roberts and Lean, 2008). The FSS is the mean-square error of the observed and forecast fractions for a neighbourhood of length n, relative to a low-skill reference forecast. Values range between 0 and 1, with higher values indicating a more skilled forecast.

## 4 Results and discussion

## 4.1 Example cases

This section presents examples that illustrate the difference of adding polarimetric variables on top of single-polarization radar data for hazard prediction purposes. However, unlike for lightning and heavy precipitation, no significant differences were observed in the hail prediction, consequently no example is provided here.

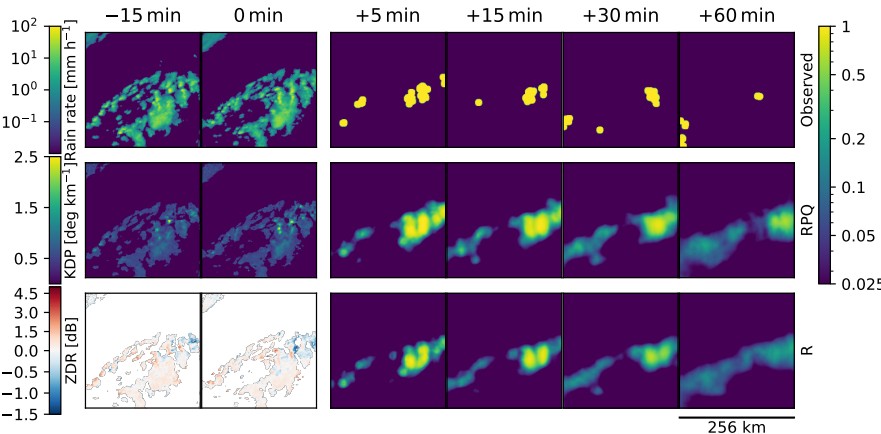

**Figure 1.** Results of the lightning prediction on 10 July 2020, 19:10 UTC. On the left three input variables are shown (Rain rate, $K_{dp}$ and $Z_{dr}$), and on the right the observed lightning occurrence and the predicted lightning probability according to the input sources RPQ and R at different lead times (indicated at the top of each column) are shown.

Figure 1 serves as an example of the model output for lightning for several time steps. This event took place on 10 July, 2020, which was characterized by a low pressure system over Scandinavia, that steered a cold front towards Switzerland. Ahead of this front, very warm and humid air flowed from southwest towards the Alps. The gradual humidification of the various layers of the atmosphere and the inflow of more unstable air first activated the diurnal cycle of showers and thunderstorms in the Alps and then the pre-frontal thunderstorm activity, particularly in southern and western Switzerland.

Both data source combinations (R: single-pol radar and RPQ: single-pol radar, polarimetric variables and quality indices) are able to accurately predict the location of the lightning (see Fig. 1). However, the difference is in the certainty of the predicted lightning over all lead times, with higher probabilities seen in RPQ. Locations where RPQ is more certain compared to R are also at locations with higher $K_{dp}$ values.

In Fig. 2 an example for the prediction of rain exceeding 10 mm is shown. This event took place on 07 June 2020 and was characterized by a low pressure area that was very pronounced throughout the depth of the troposphere, moving southward over the North Sea. A related cold front was located over the northern side of the Alps. Ahead of this front, a southwesterly flow conveyed very humid and unstable air toward the Alps; behind, the cold front colder polar maritime air moved from the northwest to the southeast. There was an strong air mass gradient with very pronounced instability in the Alps.

Figure 2 shows that both R and RPQ are able to accurately predict the location of the rainfall. However, compared to R, RPQ is more certain about the precipitation in the lower area of the rainfall field, which corresponds with the observed probability. These locations also have higher $K_{dp}$ values, which can be an indication of heavy rain.

Overall, we see similar spatial patterns in the predictions for lightning and precipitation when using RPQ compared to R, but RPQ tends to give higher confidence in the predictions.

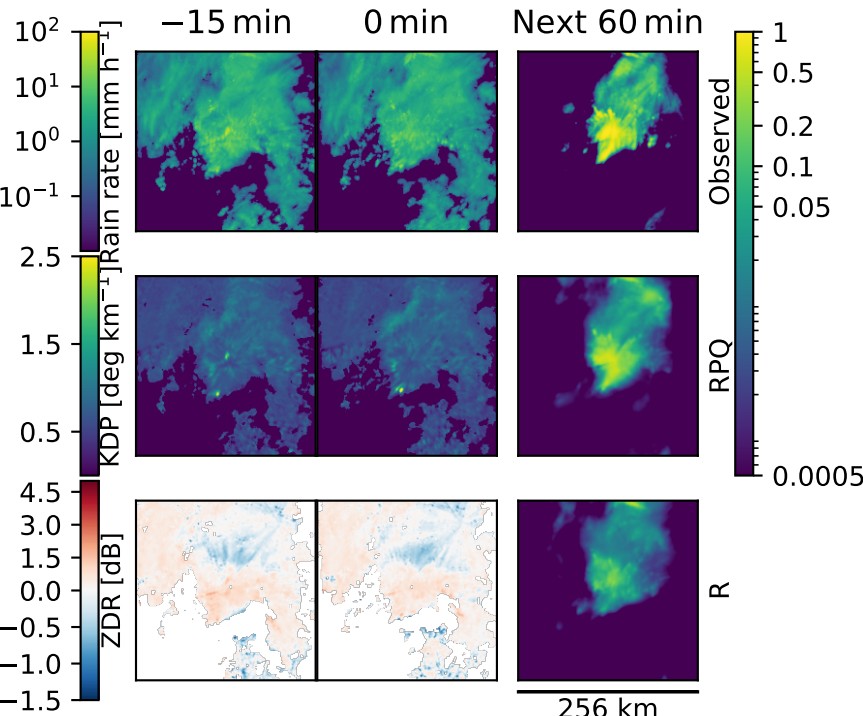

**Figure 2.** Same as Fig. 1, but for heavy precipitation on 07 June 2020, 08:50 UTC. Only one output is shown as the precipitation is predicted as the accumulation over the next 60 min.

## 4.2 Predictor importance

The average loss and the unbiased sample standard deviation, derived from the test dataset, are shown in Fig. 3a for lightning, indicating that incorporating polarimetric variables with the single-pol radar source improves the overall outcome. While including all sources (RPQ) for the lightning model results in the highest skill, it is within the spread of RP or RQ, indicating that multiple runs are necessary to verify the robustness of the results, avoiding that coincidental convergence resulted in slightly better or worse results.

Figure 3b indicates that although incorporating either polarimetric variables or quality indices on top of the single-pol radar source improves the performance for hail, surprisingly this does not hold when including all three predictor sources.

From the losses for heavy precipitation (Fig. 3c), it is evident that incorporating polarimetric variables benefits the results, and produces the most significant improvement compared to lightning and hail. While the pattern of the standard deviation are somewhat similar to that of lightning, the average loss between the model combinations lie further apart.

For lightning, the performance marginally improves from $0.335$ (using single-polarization radar and quality index) to $0.333$ when using all data sources. However, the difference is similar to the standard deviation of the losses of the three training runs. For hail, the standard deviations of the losses are even higher than for lightning (Fig. 3a) and for rain (Fig. 3c). An increase of

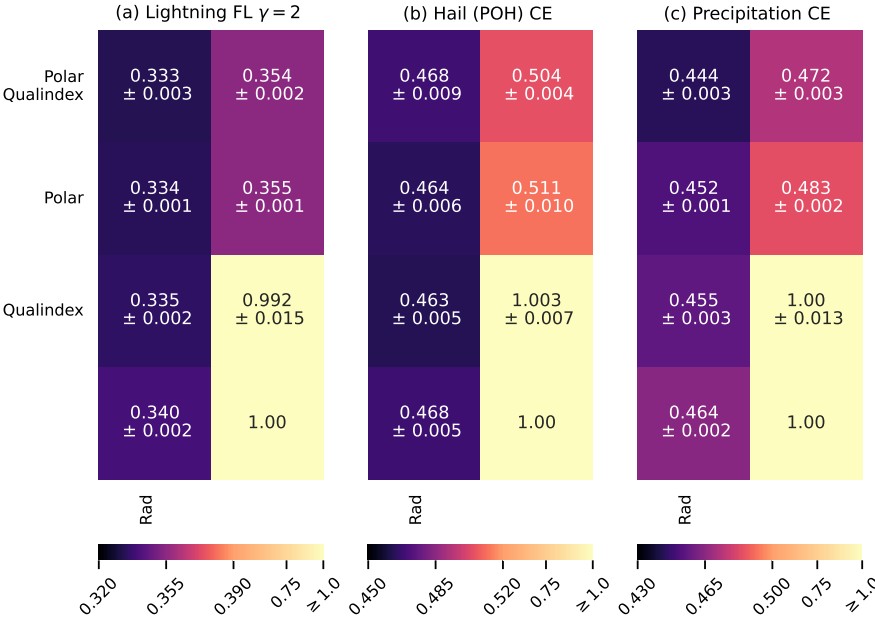

**Figure 3.** The average loss in the test dataset for the prediction of (a) lightning, (b) hail and (c) heavy precipitation using different combinations of the selected data sources. The mean loss and the spread of the three runs is shown here. Each panel shows a matrix where the data sources corresponding to each element can be found by combining the row and column labels. With "R" indicating single-pol radar data, "P" the polarimetric variables and "Q" quality indices. All loss scores are normalized with the same loss value from from Leinonen et al. (2023), such that the baseline model (model without any input) is set to 1.

the loss from $0.463$ using single-polarization radar and quality index to $0.468$ when using all three data sources is within the standard deviations of $0.005$ or $0.009$, respectively.

A reason for the larger spread of the hail results might the indirect retrieval method of the POH. While the precipitation radar and lightning sensors are designed for a direct observation of precipitation and lightning, the hail retrieval is a parametrization based on the vertical extent of the updraft core, i.e. a macroscopic property of the storm. Therefore, the POH observations - used as reference - might be less precise in comparison to precipitation and lightning observations, and, in consequence, could cause higher variation of the training performance.

As a final remark, the performance of a machine learning algorithm does not always improve when adding more predictors. In case of highly correlated or redundant predictors, no additional information content is added. However, a larger number of weights must be trained, which typically requires a larger training dataset. Furthermore, a more complex algorithm is more prone to overfitting.

**Table 1.** Normalized Shapley values in the test dataset for the input sources (R: single-pol radar, P: polarimetric variables and Q: quality indices), and the prediction of lightning, hail and heavy precipitation.

|  | R | P | Q |
|---|---|---|---|
| Lightning | 0.508 | 0.481 | 0.012 |
| Hail | 0.537 | 0.463 | 0.000 |
| Precipitation | 0.508 | 0.475 | 0.018 |

**Table 2.** Comparison of the average PR AUC and standard deviation of different model configurations (R: single-pol radar and RPQ: single-pol radar, polarimetric variables and quality indices) with the test set. For hail and lightning the average over all lead times is shown; for precipitation, the score is given for the accumulated precipitation in 1 hour exceeding 10 mm.

|  | PR AUC | |
|---|---|---|
|  | R | RPQ |
| Lightning | $0.626 \pm 0.001$ | $0.632 \pm 0.003$ |
| Hail | $0.239 \pm 0.005$ | $0.235 \pm 0.010$ |
| Rain | $0.466 \pm 0.003$ | $0.482 \pm 0.005$ |

### 4.3 Shapley values

Another method to quantify the importance of the data sources is by computing the Shapley score. This was calculated for the model runs with the optimal loss score (i.e. the model with the lowest loss out of three runs). The Shapley values for all thunderstorm hazards indicate the same: that single-polarization radar is the most important source, followed by polarimetric variables (Table 1). The single-polarization radar source is relatively more dominant for hail compared to lightning and heavy precipitation. While previous results (Fig. 3) showed that including Q improves the results, the Shapley score indicates that the importance of Q is small. However, the effect of Q is relatively independent of R and P, whereas R and P contain redundant information, and consequently, one does not add that much over the other. The Shapley value is computed from the marginal contributions of the predictors and thus does not fully capture this interdependence of features.

### 4.4 Performance of the forecasts

To get a more complete understanding of the skill of the model to predict the different variables, it is also important to see how it performs using other metrics. In Table 2 the average PR AUC and unbiased sample standard deviation are given. These values align with the loss, indicating that for both lightning and rain the model improves by incorporating all sources, with the largest improvement seen in precipitation. Meanwhile, for hail the RPQ model results in a slightly lower skill when including all sources instead of the single-polarization radar source alone. In addition, the least consistency is seen in the results of RPQ for hail.

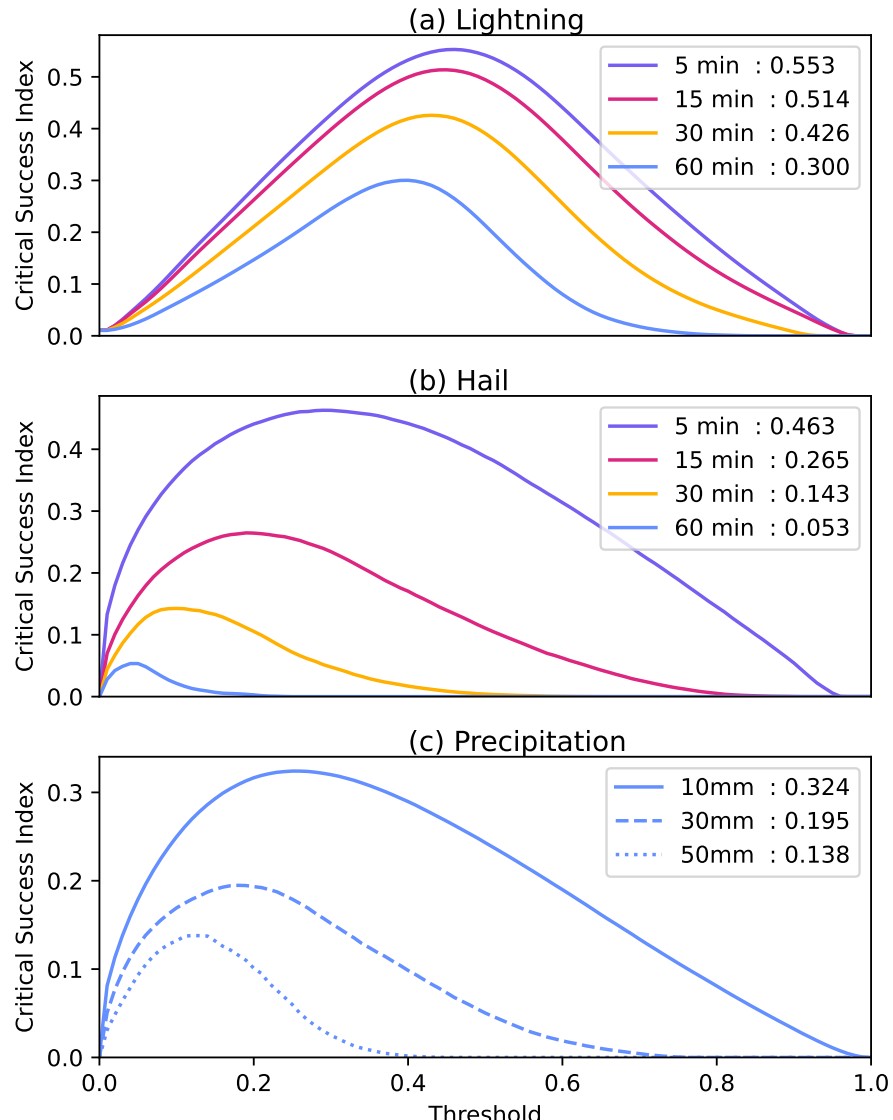

**Figure 4.** Critical Success Index over the test dataset at different thresholds for (a) lightning and (b) hail for different lead times and (c) the accumulated precipitation in 1 hour exceeding 10 mm, 30 mm or 50 mm, using the source combination "RPQ" (single-polarization radar, polarimetric variables and quality indices). The value behind the lead time or class in the legend indicates the optimal CSI.

We also investigated the effect of different probability thresholds and lead time on the skill of the forecasts. In Fig. 4 the CSI was calculated for different thresholds. For hail and lightning this was done for lead times of 5, 15, 30 and 60 min. With increasing lead times the skill of the forecasts decreases. The decrease in skill is more gradual for lightning, while for hail the values drop quickly, having barely half of the maximum CSI (indicated value in the legend) after 15 min compared to 5 min.

**Table 3.** Optimal Critical Success Index over the test dataset, calculated for different probability thresholds to transform POH to binary fields.

|  | **CSI** | | | |
| --- | --- | --- | --- | --- |
|  | 5 min | 15 min | 30 min | 60 min |
| POH $\geq 30\%$ | 0.469 | 0.263 | 0.148 | 0.057 |
| POH $\geq 50\%$ | 0.463 | 0.265 | 0.143 | 0.053 |
| POH $\geq 80\%$ | 0.420 | 0.236 | 0.108 | 0.037 |

For heavy precipitation, CSI was calculated over the accumulated precipitation in 1 hour for the three classes. Figure 4c indicates that more extreme precipitation is more difficult to predict. The lifetime of precipitation events decreases with higher rain rates, affecting the skill of the forecasts.

It is also evident that the threshold resulting in the highest CSI is not fixed over the lead times (for lightning and hail) or over the classes (for precipitation). Thresholds should be decided on by the end users, selecting values that fit their desired criteria.

In Fig. 4b the target variable POH was transformed to binary fields by considering POH $\geq 50\%$ as hail. Selecting other probabilities to convert POH into a hail event results in a different skill, as shown in Table 3. The skill of the predictions improves when smaller thresholds are selected, that is, POH $\geq 30\%$ produces the highest skill (Fig. 4b and Table 3. Lower POH thresholds (i.e. 20–50%) are often related to graupel or soft hail (Löffler-Mang et al., 2011). While according to insurance loss data, a POH threshold of 80% is related to severe hail locally (Nisi et al., 2016). These extreme events are are less frequent, and therefore, more difficult to predict.

Lower skill for precipitation and hail than for lightning can be a consequence of the time and space scales of the target variables. This difference can be enhanced due to the definition of lightning occurrence that we inherit from Leinonen et al. (2022b). This was set to the lightning occurrence within 8 km in the last 10 min, which assigns a larger spatial and temporal footprint to the lightning. Both PR AUC and CSI are sensitive to any degree of error, i.e. it compares the occurrence of an event pixel-wise, resulting in double penalization. Matching exactly high-resolution forecasts with observed small-scale features, such as thunderstorms, is rather difficult (Ebert, 2008). For that reason the FSS is calculated over multiple scales (Fig. 5). The differences between FSS for RPQ and R are marginal, especially for shorter lead times. RPQ is slightly better for predicting lightning, with increasing differences for larger lead times (Fig. 5a), which is in line with the previous results, while for hail we find the opposite result, i.e. R is slightly more accurate compared to RPQ and differences decrease at longer lead times (Fig. 5b). For precipitation RPQ results in a higher skill for warning levels of 10 mm and 30 mm, while R is better for warning levels of 50 mm.

The machine learning model learned from a dataset that was limited to one convective season. Nevertheless, the training dataset contained around a million samples. In this paper, we chose to use the same period as Leinonen et al. (2023) to make the results comparable. By providing a dataset covering more convective seasons, it is expected that skill scores of the different model versions will improve. It is not expected that the ranking of different model versions with different input dataset

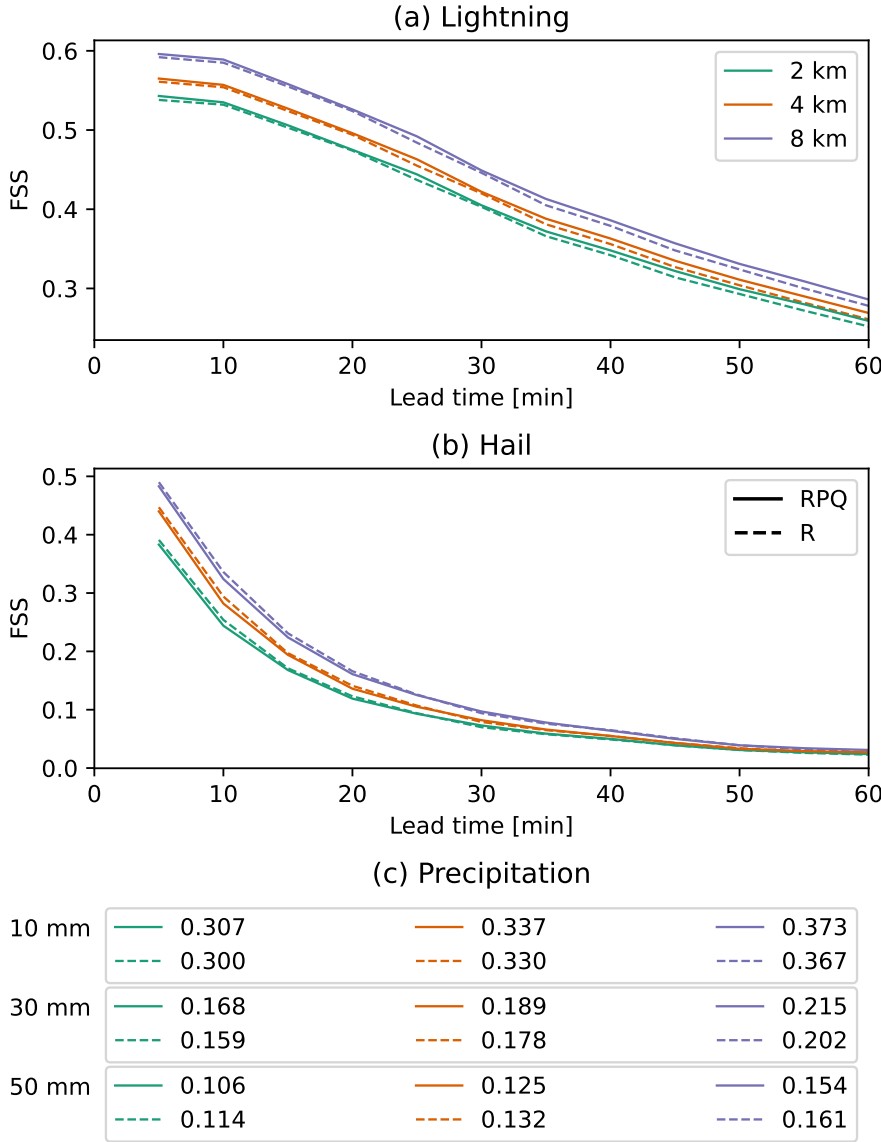

**Figure 5.** Fractions Skill Score (FSS) over the test dataset at different lead times for (a) lightning and (b) hail and (c) the accumulated precipitation in 1 hour exceeding 10 mm, 30 mm or 50 mm, using the source combination "RPQ" (single-pol radar, polarimetric variables and quality indices; solid lines) and "R" (single-pol radar; dashed lines).

will change, as more events will be available for all observation types (lightning, single polarimetric radar and polarimetric moments).

## 5 Conclusions

The objective of this work was to evaluate the benefits from including polarimetric radar observations as an additional data source for nowcasting thunderstorm hazards, compared to exploiting single-polarization radar data alone, as polarimetry provides information about the microphysical properties of hydrometeors, such as particle shape and size, consequently reducing ambiguities concerning the hydrometeor classes and drop size distribution. Additionally the benefits of exploiting radar quality indices were investigated. This work utilizes the convolutional-recurrent neural network from Leinonen et al. (2022b), which can nowcast the probability of lightning and hail occurrence up to 60 min with a 5 min resolution, as well as the probability of one-hourly accumulated precipitation exceeding pre-defined threshold levels.

The importance of the polarimetric variables (P) and quality indices (Q) is investigated by comparing model runs using extended sets of input variables compared to a reference run using only the single- polarimetric radar data (R). For all three hazards, single-pol radar is the most dominant data source according to the Shapley values. Incorporating polarimetric variables in addition to single-polarimetric radar data results in a higher skill for lightning, hail and heavy precipitation predictions. In addition, quality indices that take into account quality properties of the radar reflectivity fields have a positive impact on the results in most cases. Each model version was trained three time to test the robustness of the results. Slightly different final loss values were obtained and the standard deviation was calculated. The variation of the loss values caused by different combinations of input datasets (RP, RQ and RPQ) have a similar order of magnitude as the variations by the initial training conditions, in particular for lightning nowcasting. Differences in mean loss values should be interpreted with care and it is important to verify the robustness of the results. Among the three targets, the nowcasting for heavy precipitation improves the most when polarimetric variables are included. For hail, the results show that different input combinations are not significantly different from each other, but the differences could be rather caused by random variation within the training. Consequently, we cannot conclude that the polarimetric variables, in the form used in this study, improve the hail predictions in a statistically significant way.

Given that the nowcasting performance improve for lightning and precipitation, but not for hail, we recommend to investigate further how information of polarimetric variables, such as $Z_{DR}$-columns, can be exploited for improving hail predictions. While it is not expected that the ranking of the data importance will change, nevertheless, we recommend to include a larger training period, covering more convective seasons, in order to improve the skill of the model.

*Code and data availability.* The code used in this study can be found at https://github.com/MeteoSwiss/c4dl-polar. The datasets from the radar source are available for noncommercial use at https://doi.org/10.5281/zenodo.6802292 (Leinonen et al., 2022a). The additional datasets, models and results can be found at https://doi.org/10.5281/zenodo.7760740 (Rombeek et al., 2023).

*Author contributions.* All authors were involved in the design of the study. NR performed the data processing and analysis, with contributions from all co-authors. NR prepared the manuscript with contributions from all co-authors.

*Competing interests.* The contact author has declared that neither they nor their co-authors have any competing interests.

*Acknowledgements.* We thank Monika Feldmann for proofreading the manuscript, and Luca Nisi for providing the meteorological context of the two examples in Figs. 1 and 2. JL was supported by the fellowship "Seamless Artificially Intelligent Thunderstorm Nowcasts" from the European Organisation for the Exploitation of Meteorological Satellites (EUMETSAT). The hosting institution of this fellowship is MeteoSwiss in Switzerland.

340

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
