# Peer review of "Exploiting radar polarimetry for nowcasting thunderstorm hazards using deep learning"

_EGUsphere, 2023_

## Author Comment (AC1)

Dear reviewer,

Thank you for your time in providing detailed feedback and suggestions for our manuscript "Exploiting radar polarimetry for nowcasting thunderstorm hazards using deep learning". Below, we provide our responses to the comments of the reviewer

The Reviewer's original comments are noted below in *green italics*. Our responses are given below each comment in normal font.

Best wishes,

Authors

**General comments**

The authors present relevant and new scientific results, which fit well within the scope of the journal.

*Overall, the level of is English is good but the clarity and flow of the text can be improved, for example the following sentence is unclear: "... due to high demand in computation time results, are not available in real time." or L122": "The main difference between the predicted thunderstorm hazards is that heavy precipitation is trained ... ". Some formulations are ambiguous or confusing and should be improved e.g. L174 "The average loss of lightning".*
Thank you for pointing this out, we proofread the paper again and tried to improve the clarity of the writing; the detailed changes can be found in the attached tracked-changes file.

Based on both reviewer comments, we rewrote the second paragraph in the introduction, consequently this sentence was removed: *"... due to high demand in computation time results, are not available in real time."*

And we rewrote:
*"The main difference between the predicted thunderstorm hazards is that heavy precipitation is trained ..."*
into
*"The main difference between the predicted thunderstorm hazards is that the output of heavy precipitation is accumulated over 1 hour for predefined warning levels, whereas hail and lightning are produced at a 5-min resolution for 12 time steps (1 hour)."* (lines 158-160)

*I missed the information on how the train-test split was done (randomly, different time periods?). Did you make sure to include all kinds of events (and non-events) in both? Please provide some more details on the learning process (learning rate, stopping criterion etc.) which are crucial pieces of information for this kind of research.*

We used the same method as used in the previous research from Leinonen, but this is indeed not mentioned in the manuscript. As it is a crucial part, we added in the methods a new section about the selection of events (*section 3.1 Event selection*). We also wrote in the section 3.2 *Neural Network* (lines 173-176) about the learning rate and stopping criterion.

*I would also suggest that the authors better motivate the limited lead time of one hour. Is this enough to act upon? Or is it motivated by an inherent predictability limit to the phenomena you are trying to forecast?*

In order to make it comparable with the results of Leinonen et al. (2023) we limited ourselves also to a lead time of 60min. For lightning, the result is useful for beyond 1h. For that reason, it is possible to extend the lead times for lightning in another study. For hail, the convective nature and the small scale structure of the phenomena heavily impede the nowcasting on a local scale to a maximum useful lead time of 10 to 20min. This is rather short. In case of a fast data transfer, a user might react in time to get a benefit (e.g. move property to safety); a fully automated system like hail damage prevention for window blinds would also have time to react. An example of such a system is described at https://www.hagelschutz-einfach-automatisch.ch/eigentuemer-verwaltungen.html.

*Moreover is there a reason why rain gauges are not used as an input, or for validation of heavy rainfall?*

The nowcasting system of MeteoSwiss is designed for a very high timeliness for the reasons discussed in our response to the previous comment; we start the nowcasting 30s after the last measurement (radar data arrives in time for this). In contrast, data from rain gauges arrive in sufficient completeness around 8 min after measurements, and for the same reason CombiPrecip (product which combines the real-time radar reflectivity and rain-gauge observations) is not used an input. However, CombiPrecip is used as a target.

*It would be useful for the authors to also discuss the added value of their DL-based methodology as compared to more traditional methods in terms of computational performance. How long does it take to run this 1-hour nowcast?*

We need 8 seconds for a nowcast of one hazard with 12 timesteps on a machine with 4 CPUs (Intel(R) Xeon(R) Gold 6142 CPU @ 2.60GHz) and need 16GB of RAM. We added this to the end of subsection *3.2 Neural Network* (lines 178-179).

**Specific comments**

*L22: The sentence "the initial state of the atmosphere in NWP assimilation is based on previous model predictions rather than the latest available observations, which makes it less suited for*

*accurately predicting the time and location of convective storms": appears to stem from some misconception. Indeed, the previous forecast is used as a first guess or background field, but this is in fact combined with the latest observations to create an initial state in the data assimilation process. The main reasons why nowcasting is important is because of the faster computational time which allows a higher update frequency, and because a purely (or mostly) observation-driven system will (by construction) be closer to observations for the shortest lead times.*

Thank you for pointing out this mistake, we rewrote this paragraph into: *"NWP analysis is a combination of previous model predictions and the latest available observations, and the assimilation creates a physically consistent state of the atmosphere, which typically deviates slightly from the latest observations. Meanwhile, nowcasting algorithms aim to provide their output within tens of seconds up to a minute Pierce et al., 2012).. They typically do not strive for a physically consistent representation of the atmosphere, but do make use of the latest observations, which results in higher performance on the very short and short time scales (i.e. 1~h) and smaller scales (Simonin et al., 2017) (but inferior performance on longer lead times)..*"

*L37: The rainfall fields on which these nowcasting systems are based typically already make use of dual polarization variables to estimate the rain rate, clutter etc. What you mean is that you explicitly add polarimetric variables in the nowcasting scheme.*

In the Swiss weather radar network, dual-pol data is indeed used for clutter suppression. We included this in the *Single-pol Radar* (before named "*Weather radar"*) part in subsection *2.2 Data sources and preprocessing*.

To make it more clear that we use polarimetry variables explicitly we changed the text to: "*However, these studies primarily focus on single-polarization radar observations (e.g. precipitation rates based on horizontal reflectivity or the reflectivity itself), and do not utilize polarimetry explicitly, despite polarimetry can provide further information about the micro-physical properties of hydrometeors. Hence, adding polarimetric radar variables explicitly helps considerably to reduce ambiguities concerning the hydrometeor classes and drop size distributions .*"

*L58: "dataset from Leinonen et al. (2022b)"- can you provide a bit more detail here (which variable? Radar rainfall dataset?)*

We specified which exact dataset for training from Leinonen et al. (2022b) was used in section *2 Data*. In subsection *2.2 Data sources and preprocessing* we added the variables that are part of this dataset *Single-pol Radar* (before named "*Weather radar"*).

*L71: "Weather radar (R) observations"- please be more specific. I suppose you mean reflectivity, but there are many more weather radar observations (e.g. radial velocity…)*

We specified the corresponding variables used in the data source *Single-pol Radar* (before named "*Weather radar"*).

*L93: "After hyper-parameter turning, a value of -0.5 for was selected, as Wolfensberger et al. (2021) found that this resulted in the best parameter value" - it's unclear to me whether the*

*value selection was the result of hyperparameter tuning or just taking the best value from literature? Also, "this resulted in the best …"should probably read "this was the best…"*

The value is taken from literature. In the research of Wolfensberger et al. (2021) hyperparameter tuning was performed, which resulted in the value -0.5. We rewrote this so it is more evident that we did not do the hyperparameter tuning ourselves.

*L94: Next, the data was transformed by first normalizing it by bringing the mean close to 1 -> How was this "brought close to 1", why not simply set it to 1?*
The normalization assumes a climatologic distribution. So on average it is 1, but not for every single field. We changed it into: *"First, the polarimetric data were transformed by normalizing the standard deviation and by shifting the mean to 1."*

*L127-128: "The focal loss is an adaptation of the CE and focuses more on the difficult cases" - this is not informative, please be more specific or leave out the last part of this sentence.*
We changed this into: *"To be consistent with Leinonen et al. (2023), the focal loss (Lin et al., 2017) is used for lightning. The focal loss is an adaptation of the CE and focuses more on the pixels whose classification is more uncertain ($p_t < 0.5$). In which $p_t$ is the predicted probability of the target."*

*L129: "We trained each possible combination of data sources..." - you cannot train data (or sources, or variables), you train a model.*
This has been changed to *"We trained the model with each possible combination of data sources three times"*

*L130: trained only 3 times: is it enough?*
If you want to draw a statistically meaningful conclusion, you need to have more results. However, it is constrained due to the computational time to train all the models with all different combinations multiple times. By already training it a few times, we can get a first impression of the variance, and how robust the results are.

*L134: Shapley value, which distributes the total score among its predictors... -> In game theory it is used the score this way, but there is no "score" in the picture here, please rephrase.*
We rephrased it to:
*"The importance from individual data sources can be assessed using the Shapley value (Shapley, 1951) as a quantitative indicator of the total importance of each data source. The total contribution among the predictors is distributed by assigning a value that represents their marginal contribution."*

*L138: You convert the probabilities to binary values, why? Since both the network and POH algorithms output a probability, why not use a metric that quantifies PDF overlap/mismatch?*
We used multiple metrics to evaluate our model. The cross entropy (for POH and precipitation) and focal loss (for lightning) both also take the probability distributions into account (Figure 3).

Additionally, we did the analysis with binary values; as it was often requested by reviewers of the previous papers, we decided to add it here.

With this we mean how good the model is in predicting that the rainfall amount is exceeding the different warning thresholds (>10mm , >30mm and >50mm). To avoid confusion, we will change it to skill scores.

We changed the name of the "Weather radar" source to "*Single-pol radar".* Throughout the manuscript, where appropriate, we also changed the name from radar to single-polarization radar to better differentiate between the two different data sources.

Specified to *probability thresholds*

Corrected to double penalization

Corrected

Corrected to "*human lives"*

Corrected to "*NWP models"*

Changed to "*in recent years"*

Corrected to "*operationally available products"*

Altered to "*lie further apart"*

Changed to *"shows"*

*L232: can not -> cannot*
Corrected to "cannot"

---

## Author Comment (AC2)

Dear reviewer,

Thank you for your time in providing feedback and suggestions for our manuscript "Exploiting radar polarimetry for nowcasting thunderstorm hazards using deep learning". Below, we provide our responses to the comments of the reviewer.

The Reviewer's original comments are noted below in *green italics*. Our responses are given below each comment in normal font.

Best wishes,

Authors

**Minor revisions:**
*Some parts of the paper are difficult to understand without consulting the papers by Leinonen et al. (2002a, b, 2023). Of course, it is not useful to repeat all the details from those studies, but including the essentials would be very helpful for the reader.*
We included a new subsection about how the data is selected and split in a training, test and validation set (see *3.1 event selection*). In subsection *3.2 Neural Network* we provided more information about the learning rate, stopping criterion, and the computational performance.

*Abstract: The first five sentences are more of an introduction than an abstract; consider shortening this part and adding some more details about your specific work.*
We shortened it by removing the introductory part and wrote some more detail about the specific work.

*L18-19: NWP models are useful not only for stratiform precipitation, but also for convection. In particular, high-resolution EPS and rapidly updated cycle (RUC) models are quite good at predicting convection.*
Due to the high computational demand, it takes several tens of minutes to have the results of the NWP models available (e.g. COSMO-1E runs requires 50 min runtime). While the NWP models are quite good in predicting convection, the results are not available within tens of minutes, limiting the usefulness in the ~1$^{st}$ hour. We rewrote this paragraph (see introduction, 2$^{nd}$ paragraph), to explain this better.

*P1, last paragraph: It is unclear what is meant by "These models…are not available in real time." Besides, the general statements about NWP models do not hold true for RUC)and ensembles EPS forecasts.*
Based on this, and the feedback of the other reviewer, we reformulated this paragraph (see introduction, 2$^{nd}$ paragraph).

*L54: but also heavy rainfall? Figure 2 shows the result for nowcasting precipitation based on dual-pol variables.*

We wanted to point out that the novelty of our study is that we incorporate polarimetry also for nowcasting lightning and hail, and not only on heavy rainfall (which is also already done in the research from Pan et al. (2021)). But this is indeed not very clearly written, so we changed the sentence into: "*In addition, we investigate the potential to nowcast not only precipitation, but also hail and lightning, by utilizing polarimetric variables.*"

*Introduction: Can you describe the objectives of your study in more details? Only one sentence (L48) is too short.*
We specified that we do a data source analysis, both by performing a qualitative and quantitative analysis: "*Data source importance is explored by performing both a qualitative and quantitative analysis (i.e. focal loss or cross entropy, Shapley values, critical success index and fractions skill score).*"

*First paragraph of Section 2: A period of 5 months is very short. The affiliation of the authors suggests that they have direct access to the data. So why didn't you consider a longer period? In any case, at least in the conclusions I would expect a discussion of the reliability of the results given the short time period. Finally, please state the training period of the model and at least briefly state the data used for the model.*
We agree that using 6 months is on the shorter end. We wanted to make the results comparable with the results from the research from Leinonen et al. (2023), that is why we decided to use the same dataset. Despite the somehow short period, the dataset for training, testing and validation has a respectable size of around a million samples (not including the further diversity added by data augmentation).
We included a section about how the total training samples, how events are selected and split up in a training, test and validation set (section 3.1 event selection).

We added a small discussion at the end of the discussion section:
"*TThe machine learning model learned from a dataset that was limited to one convective season. Nevertheless, the training dataset contained around a million samples. In this paper, we chose to use the same period as Leinonen et al. (2023) to make the results comparable. By providing a dataset covering more convective seasons, it is expected that skill scores of the different model versions will improve. It is not expected that the ranking of different model versions with different input dataset will change, as more events will be available for all observation types (lightning, single polarimetric radar and polarimetric moments).*"

*L64 and Figures 1, 2: "…maximum range of observations is 246 km…" is unclear. Do you mean the study area (shown in Figures 1 and 2)? Where is the location? But why didn't you use the whole radar range? You should also explain that your study area is different from the one used by Leinonen et al. (2022). Perhaps an additional figure would help.*
We changed it to: "*The maximum observation range of a single radar is 246km*". We also wrote down the size of the study area, to make it more clear.  This is the same study area used by Leinonen et al. (2022).

*L105: Where does the 8 km distance come from? Have you performed sensitivity tests with variable distance?*

This definition is used in safety procedures at airports for takeoff and landing operations, and based on the regulations of the European Union (2017) and the International Civil Aviation Organization (2018). Hence, the choice was made to make the results of the nowcasting directly useful for this purpose. We did not perform sensitivity tests with variable distance; however, as the model is very flexible, it is very easy to change this distance and retrain it.

*L112: The classes for precipitation totals are rather coarse. Can you comment on this?*

The classes for precipitation are based on the warning levels used at MeteoSwiss. We performed an analysis with more classes for precipitation. However, this analysis showed that the skill of the model for the higher classes (corresponding to more extreme precipitation) became worse when more classes were included in total. As we focus on thunderstorm hazards, we decided to continue with the model that performed better for the more extreme cases (thus the model with only 3 classes). We added a small explanation in the text.

*Section 3.1: Could you add some more (mathematical + theoretical) details of the model used, so that a reader not familiar with CNN can get the gist?*

We included a short explanation of the purpose of the recurrent and convolutional layers and the encoder-forecaster framework: *"The recurrent connections enable to model the temporal evolution, while the convolutional connections model the spatial structure. This model has an encoder-forecaster framework, in which the encoder produces a deep representation of the atmospheric state, which is decoded into a prediction by the forecaster."*

*Section 3.2: For the interpretation of the Tables and because the Shapley score in not well known, it would be very helpful to give the range of values and their interpretation.*

We included a sentence about the interpretation of the values: *"We normalize the sum of the values of the individual components to add up to 1, with higher values indicating higher importance."*

*L138: A threshold of 50% for POH makes sense, but it would be very interesting to see how the results would change if the probability were higher (note that several studies have found a POD of ~30% for a POH of 50%, which means that POH = 50% means <40% really hail on the ground).*

There is no straightforward choice for a threshold to convert POH into hail event. E.g. car insurance loss data have verified a threshold of POH ≥ 80 % to indicate the presence of hail locally (Nisi et al., 2016; Madonna et al., 2018) for severe hail events, which is also used for the definition of hail days in the Swiss hail climatology, see
https://www.meteoswiss.admin.ch/climate/the-climate-of-switzerland/hail-climatology.html
The qualitative results are expected to be the same. Hence, we chose the threshold of 50% which is the most obvious by the mathematically definition.
We calculated the CSI again for POH, but instead for $POH \geq 30\%$ and $POH \geq 80\%$. The skill of the model improves when smaller POH thresholds are selected to convert it into a hail event (that is, $POH \geq 30\%$ gives the highest skill). These results are now shown in Table 3.

*Figures 1 and 2: Please insert the units of the color bars; it does not make sense to show ZDR in logarithmic units (values can also be negative)*

We changed both figure 1 and 2, inserted the units of the different input data used. In addition, we changed to color scale of ZDR, making negative values visible.

*L183: It's very interesting that the skill for heavy rain is increased when using polarimetric parameters, but not for hail. Are there any meteorological reasons for that (you may speculate a bit)?*

Very recent research from Martin Aregger (not yet published) indicated that ZDR columns coincides more with crowdsource data than POH. ZDR columns are often associated with the updrafts in deep moist convective storms (Kumjian et al., (2014)), and are used as a predictive tool for hail growth and may help for nowcasting where hail falls. This indicates that POH is maybe not the best ground truth for hail. Unfortunately, these ZDR columns are not yet in our database, and for that reason, we couldn't use that as a ground truth. Besides, the POH observations - used as reference - might be less precise in comparison to precipitation and lightning observations, due to the retrieval method of the POH as hail retrieval is a parametrization based on the vertical extent of the updraft core.

We added a paragraph at the end of 4.2 discussion these arguments.

*Section 4.3. Again, I miss some interpretation of the results (try to give answers or speculate about the why of the results).*

We added some interpretation of the results:

*"A reason for the larger spread of the hail results might the indirect retrieval method of the POH. While the precipitation radar and lightning sensors are designed for a direct observation of precipitation and lightning, the hail retrieval is a parametrization based on the vertical extent of the updraft core, i.e. a macroscopic property of the storm. Therefore, the POH observations - used as reference - might be less precise in comparison to precipitation and lightning observations, and, in consequence, could cause higher variation of the training performance.*

*As a final remark, the performance of a machine learning algorithm does not always improve when adding more predictors. In case of highly correlated or redundant predictors, no additional information content is added. However, a larger number of weights must be trained, which typically requires a larger training dataset. Furthermore, a more complex algorithm is more prone to overfitting."*

*"The machine learning model learned from a dataset that was limited to one convective season. Nevertheless, the training dataset contained around a million samples. In this paper, we chose to use the same period as Leinonen et al. 2023 to make the results comparable. By providing a dataset covering more convective seasons, it is expected that skill scores of the different model versions will improve. It is not expected that the ranking of different model versions with different input dataset will change, as more events will be available for all observation types (lightning, single polarimetric radar and polarimetric moments)."*

*L200: Can you specify the different thresholds considered here?*
We mean the probability thresholds here, so we wrote "probability thresholds".

*L209: "…time and space scales of the target variables" not sure on this. Lightning and hail have smaller spatial and temporal scales compared to precipitation. So I would expect a higher skill for precipitation compared to the other two.*
It is difficult to compare these scores. For lightning we use a larger range and period (8km, within the last 10min). Hail and lightning are forecasted with timesteps of 5 min, while for heavy precipitation a longer accumulation time is selected (1hour). Besides, the classes selected of precipitation are on the tail of the distribution, making it also harder to predict.

*Conclusions: This section is rather short. Consider expanding it with more substance.*
We expanded this section, wrote some more details about the method and interpretation of the results and divided it into multiple paragraphs.

**Questions/Edits/Typos:**
*L13: Not the convective storms can turn into flash floods, but the associated heavy rainfall*
We changed this into: "The heavy rainfall associated with these convective storms can turn into…"

*L15: reformulate "…by these weather phenomena"; in this sentence, these refers to flash floods (object of the last sentence); but as you know, hail in Switzerland causes the largest economic losses.*
We changed it into: "…are caused by severe weather"

*L20 "…time results…" delete results*
The comma was by misplaced, and it is now corrected to: "*Furthermore, due to high demand in computation time, results are not available in real time*"

*L21: NWP models*
Changed to NWP models

*L24-25: nowcasting is simply warning or immediate rather than early warning*
We removed "early" from this sentence

*L31: "…to take the life cycle of convective cells with growth and dissipation …"*
Corrected

*L44 and others: the term "hazard" represents the potential for harm of a certain phenomena. As this is unclear in your manuscript (no information about hail size, rain intensity), I would suggest to replace "hazard" by "phenomena" when used in conjunction with hail or precip.*
We think that the use of the word "hazard" is justified here since we focus on predicting

phenomena that have the potential to cause harm. Of our three targets, lightning is always potentially hazardous, and we specialize our model to predict heavy precipitation, which causes flash floods and landslides. Admittedly, we do not consider hail size in the models of this study, but in several studies POH values of 80% or above are used as an indicator for severe hail. Hail is also an indication of strong convective storms itself that are inherently hazardous.

*L48: delete will*
We removed "will" from this sentence, and changed it into: "… this research investigates …"

*L49: specify "model" e.g., convolutional neuronal network model*
We specified that it is a recurrent-convolutional deep learning model

*L53: I doubt whether you really retrieve relevant information about microphysics; from the dual-pol radar you can get information about hydrometeors and their characteristics and not about physics (o.k., the latter could be true, but requires complex post-processing which is not mentioned in the paper)*
We replaced "microphysics" with "hydrometeors and their characteristics"

*L59 and others: be consistent in the use of "data": either singular or plural, but do not mix.*
We changed it to plural where it was appropriate.

*L71: I would be more specific here as weather radar observations may also include polarimetric variables*
We included which specific variables are part of this data source. Besides we also included the following sentence to make clear that dual-pol data is used in the data chain (for clutter suppression): "Note that dual-pol data is used for clutter suppression in the processing chain of the Swiss operational weather radar network."

*L73: is maximum echo maximum reflectivity? And what is meant by maximum, CAPPI or the maximum in overlapping areas?*
We specified more explicitly which variables are used in the data source of radar, to make more clear what is used. The "maximum" refers to the maximum value in the vertical column.

*L87: either use (plural) or used*
Changed to used

*Eq 1: Check the dimensions of the equation. Are VIS and w dimensionless?*
We included the dimensions of the different parameters. Visibility (VIS) is in % and $\beta$ is in $m^{-1}$, so the unit of h (height above the ground level in m) is cancelled out by beta, resulting in a dimensionless w.

*L92: what is "slope of the exponential"?*

We changed the into this, it is referring to the slope of the exponent in that equation.

*L120: I suggest to refer here directly to the target values lightning, POH, and CombiPrecip*
We wrote down lightning and POH but kept heavy precipitation, because CombiPrecip provides a quantitative precipitation estimation, but we are not exactly predicting CombiPrecip. Our target is derived from CombiPrecip.

*L130: again, what are training and application period?*
The whole dataset was grouped to days, and the days were randomly assigned to training, test and validation sets. We included a section (*3.1 event selection*) about how the data is selected and split up in a training, test and validation dataset.

*L137: "ground truth" for hail is weird given the fact that POH is obtained from an integral bulk (reflectivity) only, measured aloft, and does not consider horizontal drifting between the height of the radar signal and the ground*
We replaced "ground truth" with "target variables"

*L146: "…is an imbalance…"*
Corrected

*Figures 1/2: can you write a few words about the weather situation on that day? Please indicated the date.*
We added both the dates and the meteorological context in this section.

*Figure 3: A continuous color scheme for the colorbar makes no sense here.*
We changed the color scale of the figure, to make the differences between the values more evident.

*L202: "…lead times the skill of the…" "…while for hail the values drop…"*
Added "the" in both sentences

*L205: indicats; nowcast --> predict*
Changed to indicates and predict

*L209: "…than for lightning…"*
Changed

*L2010: Reference not in brackets*
Corrected

*L2012: lightning (plural does not exist); PR, AUC, and CSI*
Changed lightning to lightning. We meant the area under the curve from the precision recall plot (not two separate things).

*L216: "…while for hail it we find…" delete "it"*

"it" is removed